# A Naturally Occurring Microhomology-Mediated Deletion of Three Genes in African Swine Fever Virus Isolated from Two Sardinian Wild Boars

**DOI:** 10.3390/v14112524

**Published:** 2022-11-14

**Authors:** Mariangela Stefania Fiori, Luca Ferretti, Antonello Di Nardo, Lele Zhao, Susanna Zinellu, Pier Paolo Angioi, Matteo Floris, Anna Maria Sechi, Stefano Denti, Stefano Cappai, Giulia Franzoni, Annalisa Oggiano, Silvia Dei Giudici

**Affiliations:** 1Department of Animal Health, Istituto Zooprofilattico Sperimentale della Sardegna, 07100 Sassari, Italy; 2Nuffield Department of Medicine, Big Data Institute and Pandemic Sciences Institute, University of Oxford, Oxford OX1 4BH, UK; 3The Pirbright Institute, Ash Road, Pirbright GU24 0NF, UK; 4Department of Biomedical Sciences, University of Sassari, 07100 Sassari, Italy; 5Osservatorio Epidemiologico Veterinario Regionale, Istituto Zooprofilattico Sperimentale della Sardegna, 09125 Cagliari, Italy

**Keywords:** ASFV, wild boar, NGS, deletion, microhomology

## Abstract

African swine fever virus (ASFV) is the etiological agent of a lethal disease of domestic pigs and wild boars. ASF threatens the pig industry worldwide due to the lack of a licensed vaccine or treatment. The disease has been endemic for more than 40 years in Sardinia (Italy), but an intense campaign pushed it close to eradication; virus circulation was last detected in wild boars in 2019. In this study, we present a genomic analysis of two ASFV strains isolated in Sardinia from two wild boars during the 2019 hunting season. Both isolates presented a deletion of 4342 base pairs near the 5′ end of the genome, encompassing the genes MGF 360-6L, X69R, and MGF 300-1L. The phylogenetic evidence suggests that the deletion recently originated within the Sardinia ecosystem and that it is most likely the result of a non-allelic homologous recombination driven by a microhomology present in most Sardinian ASFV genomes. These results represent a striking example of a genomic feature promoting the rapid evolution of structural variations and plasticity in the ASFV genome. They also raise interesting questions about the functions of the deleted genes and the potential link between the evolutionary timing of the deletion appearance and the eradication campaign.

## 1. Introduction

African swine fever (ASF) is a fatal hemorrhagic disease that affects both domestic and wild pigs and is caused by the ASF virus (ASFV) [1]. ASF was first observed in settlers’ pigs in Kenya in 1909 and was later described by Montgomery in 1921 [2]. To date, the disease is present in Africa, Europe, Asia, Oceania, and the Americas [3]. Thus, it is regarded as one of the major threats to the pig industry worldwide. There is currently no licensed vaccine or treatment available; control measures during outbreaks rely solely on stamping out affected animals, the establishment of restriction zones, and both active and passive surveillance, with subsequent massive economic losses [4,5]. Furthermore, the ability of this virus to infect different target populations, such as domestic pigs, wild boars (WBs), or other feral swine, and of being transmitted by arthropod vectors (soft ticks) [4,6] results in high morbidity and mortality, cross-border spread, and rapid diffusion at the intercontinental level [7]. ASFV is a member of the family *Asfarviridae,* and it is characterized by a large double-stranded DNA sequence. The virus genome size varies between ~170 and ~190 Kb in length and contains 160–234 open reading frames (ORFs). Based on the genetic diversity of the C-terminal region of the B646L gene, which encodes the main variable capsid protein, p72, ASFV isolates have been divided into 24 genotypes [8,9]. All genotypes are present in Africa [10], while only genotypes I and II have spread globally [10]. The ASFV strains isolated in Sardinia belong to genotype I, while genotype II is present in Eastern Europe, Asia, Oceania, and Central America [3]. Both genotypes I and II are present in China [11,12]. More recently, new incursions of ASFV genotype II isolated from wild boars were reported in January 2022 on the Italian mainland [13].

In Sardinia, ASFV genotype I was first introduced in 1978, probably as a consequence of the import of contaminated food waste [14]. Since 2015, rigorous control efforts put in place in Sardinia strongly reduced the occurrence of ASF in domestic pigs. A clear decline in both virological and serological prevalence has been proven, and no outbreaks nor PCR-positive animals have been detected in domestic or wild pig populations since April 2019 [15]. Recently, we analyzed the genetic variability and genomic evolution of 71 whole-genome sequences (WGSs) resulting from sampling over 40 years of ASF endemicity in Sardinia. Three main genetic groups characterizing three temporal waves were identified, which clearly reflected the course of the ASF epidemic without other virus introductions from outside [16]. This study further showed that the evolutionary trend of ASFV in Sardinia was generally constant and relatively slow [16], and how the genetic variability might correlate with specific human-mediated activities (such as animal movements, hunting management, and outdoor breeding) was described [15].

The hunting season in Sardinia historically took place between November and January. Starting from the 2011–2012 season, a more accurate control of the hunting activity and a more rigid regulation for hunters were put in place, leading to an increase in wild boar sampling, resulting in better epidemiological knowledge of ASFV in the wild boar population. The density distribution ranges of wild boar populations were redesigned, resulting in 11 hunting management units (HMU) with different control measures applied inside and outside each wild-boar-infected area, as established by the official ASF-EP-15-18 eradication program and subsequent modifications [15,17,18].

In this paper, we report the genetic characterization of ASF viruses isolated from two wild boars hunted in January 2019 from two different Sardinian provinces, Sassari and Nuoro. These strains are the last ASFV isolated in Sardinia. To date, natural deletions were described in ASFV genotype I isolates from Portugal (NH/P68 and OURT 88/3) [19,20] and in genotype II strains isolated in northeastern Estonia (Estonia2014) [21], Latvia (Lv17/WB/Rie1) [22], and China (HuB20, Pig/Heilongjiang/HRB1/2020) [23,24], but this is the first time that a substantial genomic deletion was identified in Sardinian ASFV isolates, despite the presence of the disease on the island for more than 40 years.

## 2. Materials and Methods

### 2.1. Ethics Statement

Two 6- to 18-month-old crossbred pigs (*Sus scrofa domesticus*) were used as blood donors. Blood samples were heparinized with 100 IU/mL sodium heparin and then used for virus isolation (described in Section 2.2) [25]. The animals’ health status was routinely monitored by trained veterinarians, and their blood samples were screened for several pathogens: ASFV [26], porcine parvovirus (PPV) [27], porcine circovirus 2 (PCV2) [28], porcine reproductive and respiratory syndrome virus (PRRSV), and Mycoplasma hyopneumoniae, the last two using commercial real-time PCR kits (LSI VetMAX™ PRRSV EU/NA and VetMAX™-Plus qPCR Master Mix, both Thermo Fisher Scientific (Waltham, MA, USA), respectively), according to the manufacturer’s instructions.

Tissue samples from wild boars were collected by veterinarians of the Italian Animal Health Service during the 2018–2019 hunting season. Animals were already dead at the time of sampling; thus, the approval of the ethics committee was not required.

Animal husbandry and handling procedures (bleeding) were performed according to the Italian Legislative Decree no. 26, dated 4 March 2014, and in agreement with the Guide for the Use of Laboratory Animals issued by the Italian Ministry of Health (authorization no. 1232/2020-PR). Animals were housed at the experiment station of the Istituto Zooprofilattico Sperimentale (IZS) of Sardinia (‘Surigheddu’, Sassari, Italy).

### 2.2. Sampling, Diagnostic Tests, and Virus Isolation

Tissues were initially tested for virus presence by real-time PCR with amplification targeting the B646L (p72) gene [26]. The presence of infectious ASFV was assessed using the Malmquist test (hemadsorption test) [25]. ASFV isolation was carried out on either homogenized spleen (7212WB/19) or lung (7303WB/19) tissues. In detail, tested samples were added to two-day-old porcine monocytes/macrophage monolayers, and cells were monitored daily for five days for the hemadsorption effect, in accordance with the WOAH Manual of Diagnostic Test and Vaccines for Terrestrial Animals [25]. When the presence of live ASFV was confirmed, culture supernatant was collected and stored at −80 °C until it was used for genome sequencing. 

Sera samples of these animals were tested for ASFV antibodies (Ab) presence using a commercial ELISA test (Ingezim PPA Compac^®^, Ingenasa, Madrid, Spain) as screening test, and then confirmed by tan Immunoblotting test (IB), in accordance with the WOAH Manual of Diagnostic Test and Vaccines for Terrestrial Animals [25].

### 2.3. DNA Extraction, PCR Assay, and Sanger Sequencing

Viral DNA was extracted directly from the cell culture supernatant using a QIAmp UltraSens Virus Kit (Qiagen, Hilden, Germany), according to the manufacturer’s instructions.

For molecular analysis, different PCRs were set up. The sequences of the B602L (genome location: bases 96,322–97,938) and EP402R genes (genome location: bases 68,928–70,112) were confirmed by Sanger sequencing using primers and the methods that were described previously [29].

Furthermore, to confirm the deletion of 4342 base pairs, we designed a set of specific primers with Primer-BLAST (Appendix A) using the following PCR protocol: 1× PCR buffer, 2.5 mM MgCl2, 0.2 mM dNTPs (Invitrogen, Thermo Fisher Scientific, Waltham, MA, USA), 0.2 µM of each primer, and 0.25 U of Platinum Taq (Invitrogen) in a total volume of 50 µL. The incubation profile was established as follows: 95 °C for 5 min, followed by 40 cycles of 95 °C for 30″, 50 °C for 30″ (primer, ASFV_4DEL, ASFV_5DEL, ASFV_6DEL, and ASFV_8DEL), 55 °C for 30″ (primer ASFV_1DEL, ASFV_2DEL, ASFV_3DEL, ASFV 7DEL), and 72 for 60″, with a final extension of 72 °C for 10 min.

The whole region encompassing the deletion (ASFVfullDel, Appendix A) was amplified using the primers ASFV_1DEL forward and ASFV_8DEL reverse using the same PCR protocol outlined above with an annealing temperature of 55 °C.

The reference strain KX354450 was used as a positive control in each PCR.

The correct size of the amplicons was verified on 2% agarose gel using the precast E-Gel EX Agarose on the E-gel power Snap (Invitrogen), according to the manufacturer’s instructions. The PCR products were purified using the ExoSap-IT PCR Product Cleanup (Applied Biosystems, Thermo Fisher Scientific), according to the manufacturer’s protocol. The templates were used for cycle sequencing reactions using Big Dye Terminator version 1.1 (Applied Biosystems, Thermo Fisher Scientific). Purified products were run on an ABI PRISM 3500 Genetic Analyzer (Applied Biosystems, Thermo Fisher Scientific). Sequences were reference-aligned using ASFV reference strains retrieved from GenBank (KX354450) in BioEdit 7.2.5 [30] and MEGA 7.0 [31,32].

### 2.4. Full-Genome Sequencing and Assembly Analysis 

For full-genome sequencing, the extracted DNA of the 7303WB/19 and 7212WB/19 viruses were prepared using a Nextera DNA Flex Library Prep Kit (Illumina Inc., San Diego, CA, USA) starting from a minimum DNA input of 50 ng. After quantification by Qubit 2.0, the DNA samples were sequenced at the AMES Group, Centro Polidiagnostico Strumentale, Napoli, Italy, using an Illumina NovaSeq 6000 (Illumina Inc., San Diego, CA, USA), according to the manufacturer’s instructions. Median coverage values of 250 and 120, respectively, were obtained. Genome data processing was performed using an in-house bioinformatic pipeline. The bcl2fastq program was used to convert the BCL files generated by the sequencing systems to standard FASTQ file formats [33] that were used to quality trim the data and remove sequencing adaptors [34]. The reads were then aligned to the pig reference genome (Sus scrofa 10.2) [35] using the bwa-mem algorithm [36]. Only reads mapping uniquely to the ASFV genome were retained and realigned using GEM [37]. Aligned bam files were sorted and indexed with SAMtools [38] and deduplicated with Picard tools [39]. To obtain high-quality variants, FreeBayes [40] was used to call variants for each sample using the KX354450 [41] sequence as a reference genome (parameters: “–ploidy 1 -X -u -m 20 -q 20 -F 0.2”). Whole-genome sequences (WGSs) were aligned using MAFFT 7.427 [42], and polymorphism positions were visually inspected using Jalview 2.10.3 B.1 software [43]. Bam files of both 7303WB/19 and 7212WB/19 were aligned with the KX354450 sequence and visually inspected with IGV 2.4.14 [44]. Genome annotation was performed in GATU [45] using KX354450 as the reference genome. The Artemis Genome browser and annotation tool (Sanger Institute) allowed the visualization of sequence features, next-generation data, and the results of analyses within the context of the sequence and were used to calculate the % G~C content. The genome sequence data generated in this study are available in the GenBank database (accession numbers ON260839 and ON260838).

### 2.5. Phylogenetic Analysis 

To investigate the evolutionary relationship of the 7303WB/19 and 7212WB/19 isolates among the historical context of the ASFV genotype I in Sardinia, a phylogenetic analysis was performed using an alignment consisting of (i) 75 WGSs generated from viruses historically isolated from Sardinia between 1978 and 2019 and (ii) 3 WGSs from viruses of European and African origins. A maximum-likelihood phylogeny was reconstructed in IQ-TREE 2.2.0 [46] by modeling the nucleotide substitution with a general time-reversible (GTR+Γ) model with empirical base frequencies. The obtained phylogeny was then time-calibrated using TreeTime 0.9.3 [47] (Appendix A).

## 3. Results

### 3.1. Isolation of 7212WB/19 and 7303WB/19 and Their Genomic Analysis

The samples analyzed in this study were collected in January 2019 from two wild boars that were hunted in two distinct WB hunting management units inside the WB-infected area. The strain 7303WB/19 was isolated from a 30-month-old male sampled in Sassari province (latitude: 40.531633; longitude: 9.132233; Pattada municipality), whereas the strain 7212WB/19 was isolated from a 30-month-old female sampled in Nuoro province (latitude: 39.891; longitude: 9.499; Lanusei municipality) (Table 1, Figure 1).

Considering an average distance of 5 km between WB home ranges [48], the distance between the areas where the two animals were sampled is 76 km (Figure 1).

Strain 7303WB/19 was isolated from the lung of an ASFV-antibody-positive (Ab+) wild boar and presented a weak PCR result (Ct = 36.2), whereas 7212WB/19 was collected from the spleen of an ASFV-antibody-negative (Ab-) wild boar and presented a strong PCR result (Ct = 19.37), as described in Table 1. Both isolates showed a hemadsorbing (HAD) phenotype with no visible differences with other Sardinian isolates collected in the last 40 years (data not shown).

The lengths of the genome sequences obtained from the ASFV 7303WB/19 and 7212WB/19 isolates were 177,417 and 177,416 bp, respectively. These sequences did not include the terminal inverted repeat KP86R, KP96L, DP93R, and DP86L genes (probably due to the difficulties in assembling low-coverage reads of these regions), with GC contents ranging from 38.78% to 38.79%, respectively.

Following annotation by GATU, we identified 228 ORFs in both 7303WB/19 and 7212WB/19, with 162 protein-encoding genes involved in virus assembly, enzymes, extracellular region parts, and viral reproduction and 66 uncharacterized reading frames (URFs).

Following NGS, the comparison of 7303WB/19 and 7212WB/19 against other Sardinian strains (Appendix A) [16,49,50] evidenced new point mutations in the intergenic regions (IG) and the replacement of the G > A base in position 142,703 (relative to KX354450), resulting in a synonymous mutation located in the region coding for the S273R gene.

### 3.2. A Deletion with Respect to the Reference ASFV Isolate from Sardinia

By aligning the NGS reads to the KX354450 reference genome, we observed that the read depth dropped significantly between 11 kb and 17 kb from the 5′ end of the genome (Figure 2). The lack of coverage in this region pointed to the presence of a deletion in the genomic sequence of both isolates. The length of the complete deletion was about 4000 base pairs (Figure 2, Appendix A); a finer resolution of the exact start and endpoint of the deletion is limited by the read length.

The deletion involved the totality of the coding regions of the genes MGF360-6L, X69R, and MGF300-1L (Appendix A).

The flanking regions of the deletion were characterized by a sequence of 48 bases that is found in all Sardinia ASFV isolates. The repeated sequences involved in this perfect microhomology corresponded to the nucleotide sequence “TGGTAATTGTACTCTATAAGTTTATAAAAATTTCAGTATATTTTTTTT” located in positions 11,774–11,821 and 16,116–16,163 with respect to the Sardinian reference genome KX354450.

These 48-base sequences are the longer identical subsequences within two regions of imperfect microhomology, as shown in Appendix A. These regions span 176 bases each, are located in positions 11,656–11,831 and 15,998–16,173, and differ from each other by only 11 bases (i.e., 6.2% divergence).

Both regions involved in this microhomology appear to be conserved among all published Sardinian isolates sampled until 2018 but not among ASFV genotype I sequences from outside Sardinia (see alignments in Appendix A).

The presence of the microhomology in correspondence to the flanking regions of the deletion suggests that the deletion arose as a result of mechanisms related to non-allelic homologous recombination (NAHR) or microhomology-mediated end-joining (MMEJ) [51]. Microhomology-mediated deletions are known outcomes of these mechanisms [52,53] and represent the most natural explanation for our findings.

We confirmed that the deletion is microhomology-mediated both by NGS and by PCR using specific primers. The sequence in the 2019 wild boar isolates contained only one copy of the 47-base sequence involved in the microhomology, and the region between the two repeated sequences was completely absent (Figure 3), as expected for a microhomology-mediated deletion.

To confirm this by NGS, we included the microhomology-mediated deletion (MMD) into the KX354450 sequence, removing all bases between 11,822 and 16,163 to obtain a new reference sequence that we denote as KX354450MMD. We realigned all reads from isolates 7303WB/19 and 7212WB/19 to this KX354450MMD reference. A manual inspection of the alignment around positions 11,775–11,821 revealed the presence of perfectly aligned reads spanning the whole sequence and the flanking regions of KX354450MMD, with no significant drop in read depth (Figure 2C), confirming the microhomology-mediated nature of the deletion. The consensus from the reads perfectly matches the sequence of KX354450MMD, implying that the breakpoints of the deletion lie within the two regions of perfect microhomology. The same local sequences were confirmed by PCR. In detail, using primers pairs from ASFV1DEL to ASFV8DEL, no amplification was evidenced in the 7303WB/19 and 7212WB/19 samples (data not shown). Instead, the PCR performed with ASFV_1DEL forward and ASFV_8DEL reverse primers produced a shorter amplicon for both isolates with respect to the reference strain KX354450, which was used as positive control (data not shown). The Sanger sequencing confirmed the deletion of 4342 base pairs in the region coding for the genes MGF 360-6L, X69R, and MGF 300-1L genes plus the neighboring intergenic regions.

### 3.3. Genetic Relatedness to other Sardinian ASFV Sequences

To better understand the origin of this deletion, we analyzed the phylogenetic relationship between the two isolates, 7303WB/19 and 7212WB/19, and previously published ASFV genotype I sequences.

We first considered the region involved in the deletion. The deletion itself was the same in the two samples from 2019, while it was not present in any of the other Sardinian samples published to date, which were sampled in a period spanning the years 1978–2018.

Both flanking regions involved in the microhomology appeared to be present and conserved among all published Sardinian isolates. The 5′ flanking region was conserved only in a few samples from genotype I from Spain and Portugal, which represent the closest outgroups of the Sardinian epidemic. Surprisingly enough, it was also conserved in genotype XX. The 3′ flanking region was conserved among Sardinian isolates and more generally among all genotype I sequences but was absent from all other genotypes. Hence, the microhomology is restricted to a subclade of ASFV genotype I from Portugal, Spain, and Sardinia (Figure 4, Appendix A). The same is true for the larger regions of imperfect microhomology (Appendix A).

A phylogenetic analysis of whole-genome Sardinian sequences showed that the two ASFV isolates containing the deletion lie in the middle of the clade formed by the Sardinian isolates and therefore clearly descended from the Sardinian epidemic. The two sequences, 7303WB/19 and 7212WB/19, were also very similar to each other, strongly suggesting that the deletion originated with a single mutational event in their ancestral lineage. Their closer ancestors lie in a minor clade containing a few recent Sardinian sequences from 2004 to 2015 and are estimated to have been circulating during late 2012 (95% HPD 2010 to 2013), which might correspond to the time window in which the deletion could have occurred (Figure 5).

## 4. Discussion

In this study, we analyzed the genomic sequences of two Sardinian ASF viruses isolated from wild boars sampled during the 2018–2019 hunting season in the ‘WB infected zone’, which was a historically endemic area of ASF. Sardinia is the oldest ASF endemic area in Europe and is characterized by a peculiar epidemiological context: the virus circulates within three diverse populations (domestic pigs, free-ranging pigs, and wild boars) without the presence of Ornithodoros ticks [54,55]. 

ASFV has been present in Sardinia since 1978, and it rapidly spread due to several factors mainly related to cultural habits. In 2015, a strict eradication plan (PE-AS15-18 and subsequent additions) was put in place: biosecurity measures were remarkably improved, and several depopulation actions against illegal pigs were carried out [49,55]. Thanks to these strict control measures, ASFV circulation dropped drastically after 40 years, leading to a reduction in the number of pigs affected and resulting in improved disease management. The ASFV genome was last detected in domestic pigs in 2018 and in wild boars in April 2019 [15,17], and antibody positivity is currently detected in <1% of tested animals [49]. The strains 7303WB/19 and 7212WB/19 are the last ASFV strains isolated in Sardinia.

In a previous study 71 ASF viruses collected in Sardinia between 1978 and 2018 were sequenced, but none of them presented significant deletions in their genomes [16,50]. No deletions were observed in strains isolated from apparently healthy Ab+ free-ranging pigs [49]. The 7303WB/19 and 7212WB/19 strains are the first Sardinian isolates in which a consistent genomic deletion was observed. Evidence of naturally occurring deletions in the ASFV genome were reported in other studies in ASFV isolates belonging to either genotype I or II, which are often associated with reduced virulence [56]. Deletions were further identified in two attenuated genotype I strains in Portugal: the NH/P68 strain, isolated from chronically infected pigs in 1968 [19], and the OURT 88/3 strain, isolated from Ornithodoros erraticus in 1988 [20]. More recently, Zani et al. (2018) reported that a genotype II ASFV strain isolated from a wild boar hunted in northeastern Estonia presented a 14,560 bp deletion from the 5′ end. This isolate (Estonia 2014) was associated with a moderately virulent phenotype with reduced lethality compared to the virulent Georgia 2007/1 [21].

To date, two naturally occurring deleted ASFV isolates were identified in China: HuB20 and Pig/Heilongjiang/HRB1/2020. The strain HuB20, isolated from a domestic pig in the Hubei province of China, presented a partial deletion of the CD2v gene and the adjacent 8CR gene [23]. In vitro, HuB20 displayed a non-hemadsorbing phenotype [23]. The Pig/Heilongjiang/HRB1/2020 strain was isolated from the spleens of pigs in Harbin [24] and also presented a non-hemadsorbing phenotype. In addition, Pig/Heilongjiang/HRB1/2020 was characterized by low virulence in vivo, persistent infection, a chronic disease course, and reduced lethality in pigs [24]. In a recent study, the genome sequences of the five “field attenuated strains” described above (OURT 88/3, NH/P68, Estonia2014, Pig/Heilongjiang/HRB1/2020, and HuB20) were analyzed and compared to related virulent ASFV isolates [56]. The deletion of EP153R and EP402R was observed to occur in four of the five field-attenuated strains that were analyzed, whereas eight different genes were simultaneously absent in three field-attenuated strains: DP60R, MGF110-2L, MGF110-4L, MGF100-1R, MGF110-9L, MGF110-12L, MGF360-6L, and MGF360-14L [56]. A major deletion of MGF110 family genes was detected in Estonia 2014, but no gene loss was found in the MGF360-10L to MGF505-3R region. On the contrary, no loss of the MGF110 gene was found in NH/P68 and OURT 88/3, except the deletion of MGF360-10L to MGF505-3R [56]. Researchers speculated that the MGF360 and MGF505 families, rather than the MGF110 family, might play a crucial role in the reduction in ASFV virulence [56].

Both Sardinian isolates investigated in this work (7303WB/19 and 7212WB/19) showed a hemadsorbing phenotype, and their sequences revealed the presence of a deletion of 4342 bases near the 5′ end, encompassing the genes MGF 360-6L, X69R, and MGF 300-1L. This deletion affects two genes belonging to the ASFV multigene families (MGFs), which are a group of genes located within the left terminal and right terminal of the ASFV genome [1]. Depending on the sizes of the MGF proteins, they can be divided into five families, including MGF-100, MGF-110, MGF-300, MGF-360, and MGF-505. Each MGF family is present in multiple copies per genome. MGF proteins differ greatly among viruses due to frequent duplications, deletions, and inversions. It has been reported that MGF proteins play important roles in multiple steps of viral infection, including transcription and translation, virulence, and immune escape [57]. For example, the deletion of the MGF-360 and MGF-505 genes have been shown to attenuate a highly virulent isolate of ASFV [58], and a different study showed that MGF-360 can suppress IFN-I responses and improve the proliferation efficiency of the virus [58,59]. Nevertheless, the function of many of these genes is still unknown. Both 7303WB/19 and 7212WB/19 presented the deletion of MGF 360-6L; future studies should investigate whether this deletion might be linked to reduced virulence or other phenotypic effects. The deletion found in this work also affected the uncharacterized X69R gene. It was first speculated that the X69R gene facilitates virus replication [60], but a more recent study demonstrated that the X69R gene is not essential for ASFV viability or its efficient replication in macrophages in vitro [61]. In addition, in vivo experiments revealed that X69R does not alter the Georgia 2007 strain’s ability to replicate or its virulence in domestic pigs [61]. Thus, it might be possible that its deletion in Sardinian ASFV isolates might not result in an attenuated phenotype. We provide high-quality evidence that the deletion is actually an MMD that originated by some processes related to recombination (NAHR or MMEJ). These processes are well-known to shape eukaryotic genomes [51,53], but they are much less studied in viruses. Our results suggest a possible wider role for MMD (and NAHR/MMEJ) in the evolution of ASFV genomes and other dsDNA viruses. In fact, recombination itself is a poorly known and poorly studied process in ASFV genomes. However, recombination can be easily overlooked when occurring between similar sequences, even in RNA viruses where intra-host recombination rates can be surprisingly large [62,63]. For ASFV, there is little direct evidence of recombination, most of it resulting from phylogenetic analyses [64,65,66], and no evidence is available related to its molecular mechanisms. Our work provides the first clear independent evidence that intra-host recombination occurs in ASFV.

Given the microhomology, this mutation had a higher chance to appear than most other structural variants. However, it appeared only once in 40 years of ASFV evolution in Sardinia. The fact that it appeared once across the whole evolutionary tree of available sequences with conserved microhomology suggests that this MMD occurs at a rate of roughly once every 350 years. This MMD has never been recorded before in any sequence from outside Sardinia, despite the fact that the microhomology is conserved across more genotype I sequences. We can speculate that either the mutation is under purifying selection or, most likely, the sampling rates for genotype I were far too low in the past to find evidence of other instances of this mutation.

It is striking that the only two sequences isolated in 2019 came from different areas of the island, yet they were both very similar and carried the mutation, hinting at a recent common origin. It is tempting to speculate that this was not a coincidence but a consequence of the fact that most of the residual viruses that were circulating in Sardinia in 2019 carried that mutation. It is also an interesting coincidence that such a mutation was observed more than 40 years after ASFV’s introduction to the island, around the time of an intense eradication campaign. Both strains were isolated from hunted wild boars. Thus, we have no information regarding the clinical status of the animals. Nevertheless, 7303WB/19 was isolated from an ASFV Ab+ wild boar with a weak PCR result (Ct = 36.2), suggesting that this animal was surviving infection. In vitro and in vivo studies will be necessary to establish if one of the phenotypic effects of this deletion is a decrease in virulence, clinical symptoms, and lethality. Attenuated ASFV strains often appeared after virulent isolates invaded a territory for a period of time, and they might be the result of the long-term coexistence and adaptation of ASFV to its host (pigs or Ornithodoros) [56]. Interestingly, the virus variant described in this study was identified more than 40 years after the introduction of ASFV to the island. Future studies should investigate whether this variant was previously circulating in Sardinia by screening other viruses collected from wild boars before 2019 in order to better understand its origin and its putative implication in the long persistence of the disease on the island.

## 5. Conclusions

This study describes for the first time the presence of Sardinian ASFV isolates with a sustain deletion in their genome (4342 bases near the 5′ end). Genomic analyses suggest that this deletion was most likely a result of a non-allelic homologous recombination driven by a microhomology. Interestingly, both strains were isolated in 2019 at the end of a rigorous eradication campaign. Our results raise questions on the functions of the deleted genes and, most importantly, whether this ASFV variant has implications in the long persistence of the disease in Sardinia or whether its appearance is linked to the successful eradication campaign that was carried out in recent years.

## Figures and Tables

**Figure 1 viruses-14-02524-f001:**
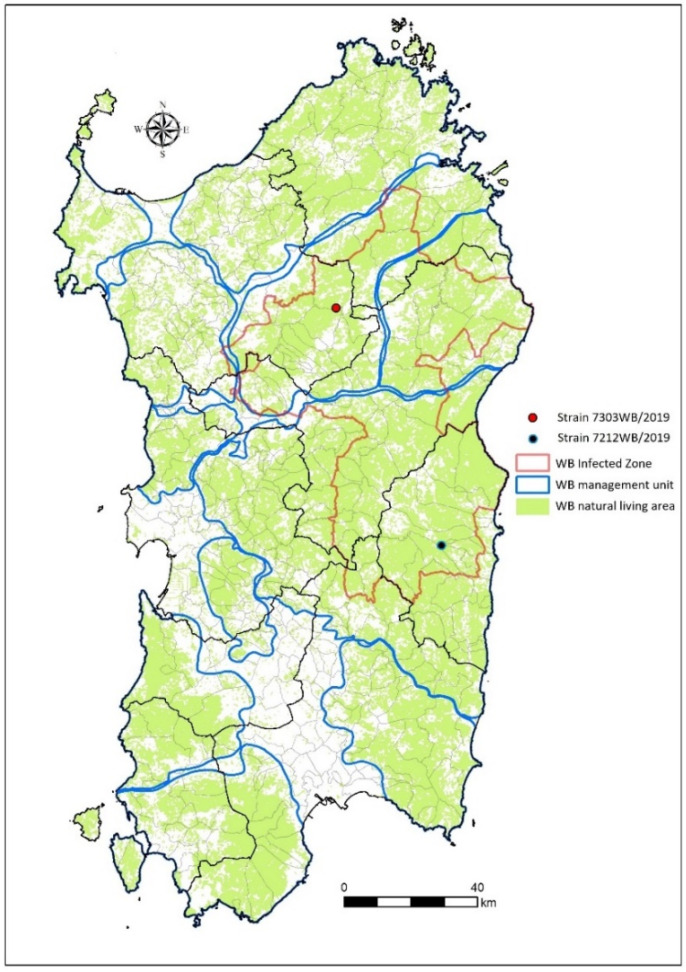
The map represents the locations of the two wild boars analyzed in this study, which were sampled in Sardinia during the 2018–2019 hunting season. The blue lines indicate the wild boar (WB) hunting management unit (HMU), while the red line indicates the limits of the WB-infected zone in 2019.

**Figure 2 viruses-14-02524-f002:**
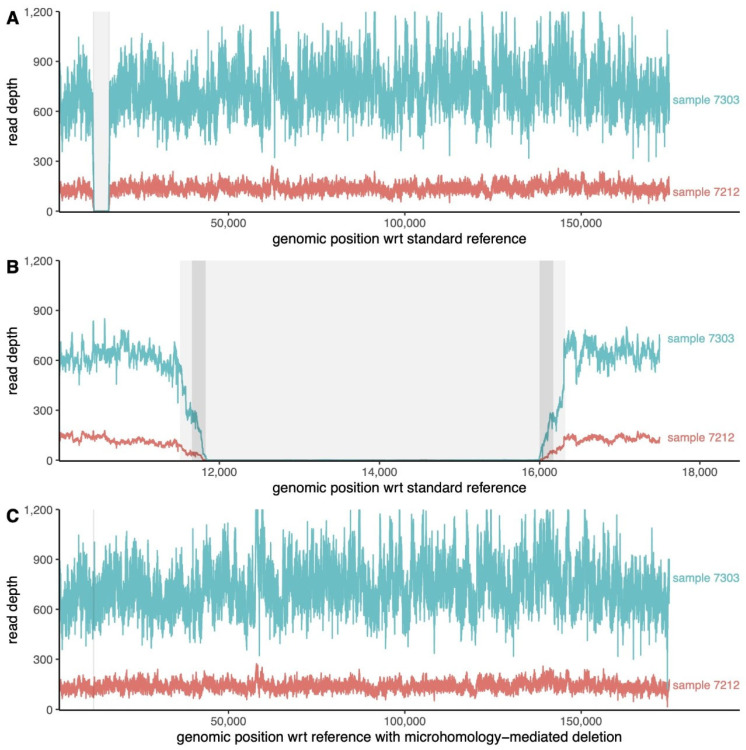
Read depth along the genome before read deduplication. (**A**) Depth of reads aligned to the Sardinian reference genome KX354450. The drop in read depth corresponding to the deletion is highlighted in grey. (**B**) Zoomed view centered on the region of the deletion (about 12–16 k from 5′ end of the genome). (**C**) Depth of reads aligned to the modified reference KX354450MMD, which included the theoretical microhomology-mediated deletion.

**Figure 3 viruses-14-02524-f003:**
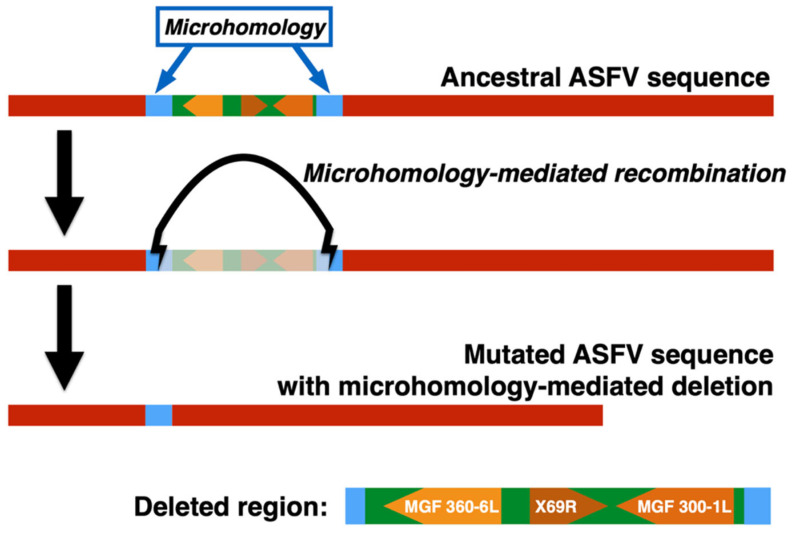
Scheme of the process of microhomology-mediated deletion and the structures of the ancestral and mutated sequences.

**Figure 4 viruses-14-02524-f004:**
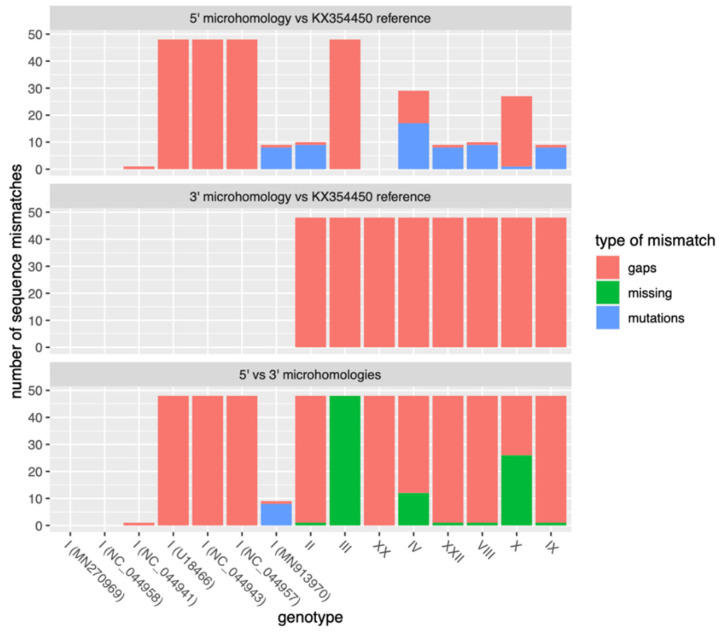
Conservation of the perfect microhomology and the two involved sequences of 48 bp (one at the 5′ end of the deletion and one at the 3′ end) among ASFV genotypes. Top: number of mismatches between the sequence closer to the 5′ end of the genome in KX354450 and the corresponding sequences in other samples from different genotypes, ordered by genomic divergence from KX354450. Middle: mismatches between the sequence closer to the 3′ end of the genome in KX354450 and the corresponding sequences in other samples from different genotypes. Bottom: mismatches between the two sequences from the same sample. “Missing” denotes bases that were missing (i.e., contained gaps in the alignment) for both sequences.

**Figure 5 viruses-14-02524-f005:**
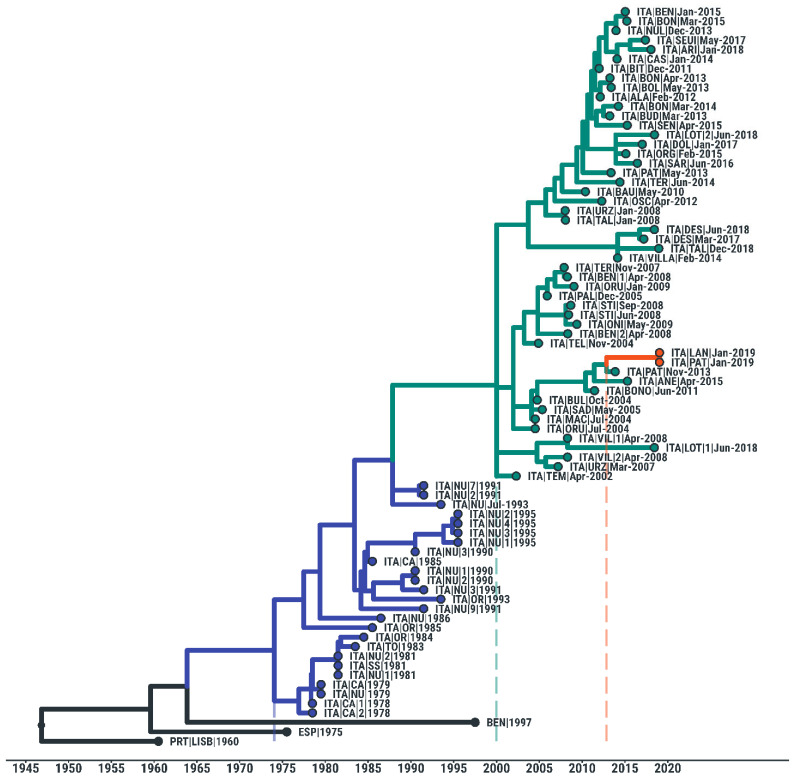
Time-scaled phylogeny of 75 ASF viruses isolated from Sardinia between 1978 and 2019. Colored branches indicate phylogenetic clades that include: (i) ASF viruses collected between 1978 and 1995 (blue), (ii) viruses isolated after 2000 (green), and (iii) the two 2019 ASFV isolates with the microhomology-mediated deletion (orange).

**Table 1 viruses-14-02524-t001:** ASF virus strains isolated from wild boar hunted in Sardinia during January 2019. Results of both virological (tested organs, Ct values of real-time PCR, and Malmquist test) and serological (ELISA and immunoblotting) tests are provided.

Strain ID °	Hunting Time (Year and Month)	Municipality (Province)	HMU ^§^	Host Species	Genotype	Organ	Ct Value *	Malmquist Test	Elisa	IB ^#^
7303WB/19	January 2019	Pattada (Sassari)	Goceano-Gallura	Wild Boar	I	Lung	36.2	pos	pos	pos
7212WB/19	January 2019	Lanusei (Nuoro)	Gennargentu-Ogliastra	Wild Boar	I	Spleen	19.37	pos	neg	neg

° ID: identification number; ^§^ HMU: wild boar hunting management unit; * Ct: threshold cycle; ^#^ IB: immunoblotting; pos: positive; neg: negative.

## Data Availability

The ASFV sequences generated during this study were deposited in GenBank. Further data presented in this study are available on request from the corresponding author.

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
