# Peer review of "A Naturally Occurring Microhomology-Mediated Deletion of Three Genes in African Swine Fever Virus Isolated from Two Sardinian Wild Boars"

_viruses, 2022, doi:10.3390/v14112524_

Round 1

Reviewer 1 Report

This paper focuses on the two ASFV isolates of 2019 from Sardinia  and is an excellent epidemiological study regarding ASF eradication and corresponding surveilance. It is amazing that there had been no variation in the circulating ASFVs during the passed 40 years before the appearance of these two isolates, probably reflecting the stability of genome organization of the ancestor/reference strain in different hosts. The typical 48-bp sequences in the ASFV isolates from Sardinia and its naturally homologous recombination resulted in a significant deletion are the great importance of this manuscript.

The paper would be better if the authors could provide some information about the clinical or pathological/necropsy changes of the wild boars carring/infecting the 7303 and 7212, although they were both obtained during hunting campaigns. Or alternatively a brief discussion about this should be given.      

Author Response

Response to Reviewer 1 Comments

  • Point 1: The paper would be better if the authors could provide some information about the clinical or pathological/necropsy changes of the wild boars carring/infecting the 7303 and 7212, although they were both obtained during hunting campaigns. Or alternatively a brief discussion about this should be given.     

  • Response 1: We thank the reviewer for his/her positive feedback. Both strains were isolated from hunted wild boar, thus no one information about clinical signs of these two animals were available.  Nevertheless, 7303WB/19 was isolated from an ASFV Ab+ wild boar, with a weak PCR result (Ct = 36,2), suggesting that this animal was surviving infection.. Future in vitro and in vivo studies will be necessary to establish if one of the phenotypic effects of this deletion would be a decrease in virulence, clinical symptoms and lethality. Text was modified at line 452-455.

A mistake was also corrected at line 236-237: the length of the genome sequences obtained from the ASFV 7303WB/19 and 7212WB/19 isolates were 177417 and 177416 bp, respectively, instead of 177594 177539.

We would like to thank the reviewer for the time spent reviewing our work.

Reviewer 2 Report

The study of molecular epidemiology of African swine fever is of great significance for the pathogenic analysis of the disease.

2020, Torresi, et al., sequenced and comparative analysed the evolution of African swine fever virus in Sardinia (1978–2014) , but they did not find significant changes. However, the study found that Sardinia had natural recombination and deletion strains, which is in line with the evolutionary trend and corresponding theories of the virus. The relevant data enrich the genome information of the African swine fever, which is of certain significance.

But there are several minor issues that need to be improved:

1 7212/19WB and 7303/19WB is inconsistent in the whole document, such problems need to be verified and corrected;

2 Figure 1 is relatively simple and useless, either delete or add detailed NTs sites data for reader reference;

3 In this paper, at least one genome of published strain should be selected as the reference strain, and the nucleotide, amino acid and ORF mutations, deletions and changes between the new isolated strains and the reference strain, and between the two new isolated strains should be listed in detail, so as to facilitate the strain to use the published data.
